# An Overview of ICA/BSS-Based Application to Alzheimer’s Brain Signal Processing

**DOI:** 10.3390/biomedicines9040386

**Published:** 2021-04-06

**Authors:** Wenlu Yang, Alexander Pilozzi, Xudong Huang

**Affiliations:** 1Department of Electrical Engineering, Information Engineering College, Shanghai Maritime University, Shanghai 200135, China; wlyang@shmtu.edu.cn; 2Department of Psychiatry, Massachusetts General Hospital and Harvard Medical School, Charlestown, MA 02129, USA; apilozzi@mgh.harvard.edu

**Keywords:** Alzheimer’s disease, blind signal separation, brain signal processing, fMRI, independent component analysis, MRI

## Abstract

Alzheimer’s disease (AD) is by far the most common cause of dementia associated with aging. Early and accurate diagnosis of AD and ability to track progression of the disease is increasingly important as potential disease-modifying therapies move through clinical trials. With the advent of biomedical techniques, such as computerized tomography (CT), electroencephalography (EEG), magnetoencephalography (MEG), positron emission tomography (PET), magnetic resonance imaging (MRI), and functional magnetic resonance imaging (fMRI), large amounts of data from Alzheimer’s patients have been acquired and processed from which AD-related information or “signals” can be assessed for AD diagnosis. It remains unknown how best to mine complex information from these brain signals to aid in early diagnosis of AD. An increasingly popular technique for processing brain signals is independent component analysis or blind source separation (ICA/BSS) that separates blindly observed signals into original signals that are as independent as possible. This overview focuses on ICA/BSS-based applications to AD brain signal processing.

## 1. Introduction

Alzheimer’s disease (AD), which was first recognized by Alois Alzheimer in 1906, is the most common cause of dementia in older adults [1,2]. According to the 2008 Alzheimer’s Disease Facts and Figures [3], recently released by the Alzheimer’s Association, an estimated 5.8 million people in the United States presently have AD, with projections indicating growth to as many as 13.8 million by mid-century. Understanding all behavioral, anatomical, and physiological aspects of this disease is vitally important to populations worldwide. Improving the accuracy of diagnosis of AD at its early stage is critical to finding a successful treatment. 

AD has a presymptomatic phase, likely lasting years, during which neuronal degeneration is occurring but clinical symptoms have not yet appeared. Critical to the early treatment of AD is the ability to discriminate between older individuals who will and will not ultimately develop the disease during this preclinical stage. Early treatment is beneficial to prevent or at least slow down the onset of the clinical manifestations of disease [4]. Moreover, to aid in the development of these treatments, specifically drugs for the treatment of AD at its early stage, early diagnostic tools and techniques to monitor disease progression in the presymptomatic phases of the disease are needed.

With the advent of biomedical engineering techniques, more and more brain signals have been acquired and processed for AD diagnosis. These brain signals come from electroencephalography (EEG) [5], magnetoencephalography (MEG) [6], computerized tomography (CT) [7], single photon emission computed tomography (SPECT) [8], positron emission tomography (PET) [9], magnetic resonance imaging (MRI) [10], and functional magnetic resonance imaging (fMRI) [11]. 

Correspondingly, a number of signal processing approaches have been proposed to process these brain signals for diagnosing patients with AD. These include: wavelet transformation (WT) [12], principal component analysis (PCA) [13], independent component analysis (ICA) [14], also known as blind source separation (BSS), parallel factor analysis (PARAFAC) [15], and so on. In this review, we will focus on applications of ICA/BSS/in processing of brain signals for potential AD diagnosis.

## 2. Alzheimer’s Disease

Since the end of the 19th and the beginning of the 20th century, it has been recognized that two classic silver-positive lesions occur in the Alzheimer’s brain: the intraneuronal neurofibrillary tangles and extracellular deposits of amyloid called senile plaques [16]. Plaques are made primarily of the amyloid β peptide, Aβ-40, and Aβ-42, derived from the amyloid precursor protein [17]. Neurofibrillary tangles, by contrast, are stained by β- pleated sheet histological reagents but are made of a different proteinaceous substance. These intracellular lesions are composed primarily of the microtubular-associated protein, tau, which changes from its normal neuronal localization in axons to occupy and dominate the somatic and dendritic compartments of a subpopulation of the large projection neurons in neocortex and limbic systems [18]. Gomez-Isla et al. introduced the concept of a third type of lesion, focusing on the neuritic changes that occur in dendrites and axons that are not (necessarily) silver-positive, but which reflect changes in morphology, trajectories, and post-synaptic structures that may also contribute to the breakdown of neural system function.

Clinical features of AD include a decline in memory and other cognitive functions. AD is the most common form of dementia, with prominent symptoms of memory loss, language difficulties (aphasia, anomia), and deficits in visual spatial skills (agnosia) and motor spatial skills (apraxia) [1]. General complaints of visual problems (difficulty reading, blurry vision, vague complaints of poor visual acuity) may be the presenting symptoms in AD, but patients may lack objective signs on ophthalmological examination [19,20,21]. Most research suggests that visual dysfunction is caused by neuropathology of the visual association cortices [22], rather than by changes in the retina, optic nerve or retino-calcarine pathways [23,24].

## 3. Biomedical Techniques for Detecting Alzheimer’s Brain Signals

With advances in biomedical techniques, it is possible to obtain Alzheimer’s brain signals to diagnose AD at an early stage. In this section, we will briefly introduce these biomedical techniques.

EEG [5] is a test used to detect abnormalities related to the electrical activity of the brain. Small metal discs with thin wires (electrodes) are placed on the scalp that sends signals to a computer for their recording. Normal electrical activity in the brain makes a recognizable pattern. Through an EEG, doctors can look for abnormal patterns that indicate seizures and other problems. EEG has good temporal resolution but relatively poor spatial resolution [25]. Since Hans Berger in 1931 first observed pathological EEG sequences in a historically verified AD patient, a large number of studies about the EEG of AD have been presented [26,27,28,29]. The primary EEG markers of AD include decreased alpha and beta activities, slower dominant-posterior rhythms, increased diffuse-slow activity, as well as a decrease in coherence [5].

MEG [30] is a complex reference-free non-invasive function brain imaging technique with millisecond temporal resolution that detects neuromagnetic signals produced by neuronal activity in the cortex of the brain. Compared to EEG, MEG is less distorted by the resistive properties of the skull, skin, and cerebral fluid, which act as a low pass filter. Several studies have been performed on AD and mild cognitive impairment (MCI) using MEG [31,32,33,34,35,36,37,38,39,40,41,42,43].

PET [9] and SPECT are two molecular imaging techniques that provide pictures of the brain that reflect the distribution of radioactive-labeled drugs (radioligands or tracers) injected into the body. Then, sensors that surround the injected body part detect the positrons emitted from the tracer in opposite directions, localizing the tracer. The most common tracer used is a glucose analogue, fluorodeoxyglucose (FDG), which measures regional cerebral glucose metabolism, a sign of neuronal activity. PET has been explored to determine its ability to differentiate between a diagnosis of AD and fronto-temporal dementia (FTD) [9,44,45,46,47,48]. SPECT also has been used to investigate functional alteration of the brain in patients with AD [46,49]. PET measurements of cerebral glucose metabolism also have been found to have superior accuracy compared to SPECT measurements of cerebral perfusion in differentiating AD from vascular dementia, regardless of dementia severity [50]. Klunk et al. presented the first human study of a novel amyloid-PET tracer, termed Pittsburgh Compound-B (PIB), in patients with diagnosed mild AD and controls and suggested that PIB can provide quantitative information on amyloid deposits [51] and detect cerebrovascular β-amyloid for identifying the extent of cerebral amyloid angiopathy (CAA) [52] in living subjects. More recent PET tracers have been found to be more effective than PIB for AD diagnosis, however [48].

CT and MRI are two biomedical imaging techniques that search for atrophy in the brain structure in vivo. With the use of CT, atrophy of medial temporal regions where AD pathology is seen early in the disease has been observed [7]. MRI techniques sensitive to changes in cerebral blood flow and blood oxygenation were developed by high-speed echo planar imaging. They allow one to obtain completely non-invasive tomographic maps of human brain activity through the use of visual and motor stimulus paradigms [53,54]. MRI has more recently surpassed CT in AD studies due to its greater accuracy, manipulability, and precision [55]. Moreover, compared to PET, MRI has the advantage of not using radioactively-labeled compounds and being non-invasive and safe for repeat studies. For AD diagnosis, MRI can act as a sensitive tool, detecting structural brain abnormalities, consistently revealing atrophy of hippocampus [56], entorhinal [10], and temporal–parietal cortices [57].

As discussed by Pekar [54], fMRI provides the opportunity to study brain function non-invasively. Since the early 1990s, it has been a powerful tool used in both research and clinical arenas [53]. The most popular form of fMRI uses blood-oxygenation-level-dependent (BOLD) contrast, that is based on the differing magnetic properties of oxygenated (diamagnetic) and deoxygenated (paramagnetic) blood. When brain neurons are activated, a localized change in blood flow and oxygenation results that causes a shift in the magnetic resonance (MR) decay parameter T2*. These blood flow and oxygenation (vascular or hemodynamic) changes are temporally delayed relative to the neural firing, a confounding factor known as hemodynamic lag. Although fMRI does not share the temporal resolution of EEG or MEG, it does have a spatial resolution of millimeters, and the experiments suggest that it may detect activations at the level of the cortical layers [58].

The detection of changes in neural activity using BOLD-fMRI [53,59] generally involves the identification of voxel signals that correlate with an imposed experimental paradigm [60].

## 4. Theory and Model of ICA/BSS

The essential problem of blind source separation is the isolation of original signals from their resulting mixture that is gathered from an array of sensors, without any information about the original signals or how they are mixed. BSS can be applied to a variety of fields, including audio processing, and is not unique to neuroimaging [61,62].

A fairly general BSS problem can be formulated as:*y*(*k*) = W*x*(*k*)(1)
where *x*(*k*) = [*x_1_*(*k*), *x_2_*(*k*), …, *x_m_*(*k*)]*^T^* is the observed sensor signals, W is the unmixing matrix, *y*(*k*) = [*y_1_*(*k*), *y_2_*(*k*), …, *y_n_*(*k*)]*^T^* is the ouput signal, and *k* is a discrete time. BSS assumes that *x*(*k*) is the output signal from an unknown and inverse multiple-input/multiple-output (MIMO) mixing and filtering system, *x*(*k*) = A*s*(*k*) (A is the mixing matrix), in which the inputs are the source signals *s*(*k*) = [*s_1_*(*k*), *s_2_*(*k*), …, *s_n_*(*k*)]*^T^*. It should be noted that *n* will be less than *m* if the system is not inversed.

The objective of BSS is to estimate the original source signals *s*(*k*). To separate the source signals *s*(*k*), a number of approaches have been developed such as ICA and its extensions, sparse component analysis (SCA), sparse principal component analysis (SPCA), non-negative matrix factorization (NMF), parallel factor analysis (PARAFAC), and so on.

PCA [63], one of well-known unsupervised analysis methods, projects the data into a new space spanned by the principal components. Each successive principal component is selected to be orthonormal to the previous ones and to capture the maximum variance that is not already present in the previous components.

ICA [64,65], as a specific embodiment of BSS, has been further developed in the last few decades. ICA, [66,67,68] as a generalization of PCA, separates the observed signals into statistically independent components using higher-order statistics whereas PCA obtains uncorrelated components using only second-order statistics. ICA is different from BSS; The basic goal of ICA is to solve the BSS problem by expressing a set of random variables (observations) as linear combinations of statistically independent component variables (source signals), whereas the objective of BSS is to estimate the original source signals, even if they are not all statistically independent. A more thorough description of ICA has been written by Comon (Comon, 1994 #58).

The technique of ICA was first used in 1982 for analyzing a problem pertaining to neurophysiology [69]. In the middle of the 1990s, after the term ICA was first coined by Comon [70], ICA received wide attention and growing interest when many efficient approaches were put forward, such as the Informax principle by Bell and Sejnowski [64], natural gradient-based infomax by Amri [71], and the fixed-point (FastICA) algorithm by Hyvärinen [72,73].

ICA [64,65] is becoming increasingly popular as a tool for analyzing biomedical data [14,74,75,76]. In the next section, we will introduce the basic model of ICA/BSS of fMRI and its variants for different purposes of application.

## 5. ICA/BSS Model for fMRI

Due to the spatial and temporal natures of fMRI data, the use of ICA can generally be grouped into two groups, namely, spatial ICA (sICA) and temporal ICA (tICA). These techniques are discussed in general terms, and the studies do not pertain to AD (or another specific neurological disorder) unless otherwise stated.

### 5.1. Spatial and Temporal ICA Models of fMRI

Makeig et al. [14] have first applied spatial instantaneous mixing ICA to the analysis of EEG data and event-related potential (ERP) data using the original Infomax algorithm [64]. Independently, Vigario et al. [77,78] have developed a method for artifact identification and noise removal from EEG and MEG through a FastICA algorithm [72].

Since then, ICA has become increasingly popular for analyzing biomedical data [14,74,75,79], especially for analysis of biomedical imaging, such as fMRI data [80]. A typical model for applying ICA to fMRI data, was introduced in a study [81] and provided a framework for understanding ICA as it applies to fMRI data and for introducing the various processing stages in ICA of fMRI data.

As an example, in the ICA processing of fMRI time series data by Calhoun et al, data was generated from a set of statistically independent (magnetic) hemodynamic source locations in the brain. These sources have weights that specify the contribution of each source plied by each source’s hemodynamic time course [80,81]. The first stage of the data generation takes place within the brain, in which the sources are mixed. The second stage of data generation involves the fMRI scanner. The sources are sampled, and each represents a function of scan-specific MR parameters such as flip angle, slice thickness, pulse sequence, and field-of-view.

Data preprocessing consists of a number of possible preprocessing stages, including slice phase correction, motion correction, spatial normalization, and smoothing. After preprocessing, it is common to perform data reduction such as dimensionality-reductions using PCA or some other approach. The resultant estimated source, along with the unmixing matrix, can then be thresholded and presented as fMRI activation images and fMRI time courses, respectively [80,81].

Conventional ICA embodies the assumption that data can be decomposed into underlying sources that are independent over space (spatial) [79] or time (temporal) [14,82] and that the probability density functions (pdf) of these sources are highly kurtotic (distribution has heavy-tails) and symmetric [79,82,83]. Different assumptions can be made between sICA and tICA [82]; sICA seeks a set of mutually independent component (IC) source images and a corresponding (dual) set of unconstrained time courses [79]. By contrast, tICA seeks a set of IC source time courses and a corresponding (dual) set of unconstrained images [14]. In concrete fMRI data, sICA finds independent images and a corresponding set of dual unconstrained time courses and embodies the assumption that each image in **X** is composed of a linear combination of spatially and statistically independent images. Unlike sICA, tICA finds independent time courses and a corresponding set of dual unconstrained images and embodies the assumption that each eigensequence in **X** is a linear combination of temporally and statistically independent sequence **S** [83].

In addition, spatiotemporal ICA (stICA) [83] embodies the assumption that each eigenimage in ***A*** is a linear combination of spatially independent images, and each eigensequence in ***S*** is a linear combination of temporally independent sequences.

### 5.2. Variants of ICA Models for fMRI Data

Moreover, researchers have proposed many variants of ICA/BSS based on differently statistical characteristics [83,84] in fMRI data or on different purposes of analyzing fMRI data [15,85,86]. The variants include probabilistic ICA, skew-ICA, group ICA, tensor ICA, and cortex-based ICA.

***Probabilistic ICA***. To address the issues of what is attributable to the “real effects” of interest and what simply is due to observational noise, Beckmann et al. examined the probabilistic ICA (PICA) model [84,87] for fMRI data. PICA allows for a nonsquare mixing process and assumes that the data are confounded by additive Gaussian noise.

***Skew-ICA***. Stone et al. [83] combined spatiotemporal ICA and skew-ICA, to form skew-stICA to analyze synthetic data and data from an event-related, left-right visual hemifield fMRI experiment. Results [83] obtained with skew-stICA are superior to those of PCA, sICA, tICA, stICA, and skew-sICA. Here, skew-ICA is based on the assumption that images have skewed pdfs [88], an assumption consistent with spatially localized regions of activity. By contrast, conventional ICA is based on the physiologically unrealistic assumption that images have symmetric pdfs.

The skew-pdf can be described as: (2)$$p(y;a,b)\propto \exp\left(\frac{a-b}{2}y-\frac{a+b}{2}\sqrt{y^{2}+1}\right)$$
where the constants *a* and *b* define the skewness of the distribution.

***Group ICA***. ICA has been successfully utilized to analyze single-subject fMRI data sets and extended to group ICA for multi-subject analysis [15,85,86,89]. Group analysis of fMRI is important to study specific clinical and experimental conditions within or between groups of subjects [90].

Calhoun et al. [85] proposed a group ICA model that was a novel approach for drawing group inferences using ICA of fMRI data. The group ICA analysis revealed task-related components in the left and right visual cortex, a transiently task-related component in bilateral occipital/parietal cortex, and a non-task-related component in bilateral visual association cortex. The group ICA approach had been implemented in the Group ICA for fMRI data (GIFT) [91], in which PCA is used to whiten the data by performing an orthogonal transformation and to reduce the number of principal component present in the mixture.

Svensen et al. [86] used the extended ICA of fMRI data from single subjects to simultaneous analysis of data from a group of subjects. The results demonstrated that group ICA can extract nontrivial task-related components without any *a priori* information about the fMRI experiment and identify components common to the whole group in analysis of group data.

***Tensor ICA***. Beckmann et al. extended the single-session probabilistic ICA [84] to a higher dimension, called tensor PICA, to analyze multisubject or multisession fMRI data. The tensor PICA [15] is derived from parallel factor analysis (PARAFAC) [92] and has three-way, including temporal, spatial, and subject-dependent, variations. Real fMRI activation data was decomposed by the approach to extract plausible activation maps, time courses, and session/subject modes that give simple and useful representations of multisubject or multisession fMRI data.

***Cortex-based ICA***. Cortex-based ICA (cbICA) assumes that cortical data are different from non-cortical data and processes a subset of the data determined by *a priori* information [93]. Formisano et al. used the mesh of the white matter/gray matter boundary, automatically reconstructed from high-spatial-resolution anatomical MR images, to limit the sICA decomposition of a coregistered functional time series to those voxels that are within a specified region with respect to the cortical sheet.

Comparisons between cbICA and other methods showed that cortical surface maps and component time courses blindly obtained with cbICA reliably reflect task-related spatiotemporal activation patterns and that the cbICA improves the fitting of the ICA model in the gray matter voxels, the separation of cortical components, and the estimation of their time courses, particularly in the case of fMRI data sets with a complex spatiotemporal statistical structure.

## 6. ICA/BSS Applications to Brain Signal Processing for AD Diagnosis

In this section, we will present some problems associated with ICA/BSS applications to brain signal processing for AD diagnosis, such as why ICA/BSS is successful when applied to the diagnosis of AD, what are useful components and noises, how many components should be extracted, and which algorithms of ICA/BSS are most suitable. Applications of ICA to the development of machine learning models will also be discussed.

### 6.1. Why Apply ICA to Diagnosis of AD

The simplest and most robust technique for analyzing MRI brain scans is region-of-interest (ROI) analysis [94]. ROI [95,96] analysis of the brain structure is considered the gold standard against which new techniques are compared, but it has some drawbacks such as operator-dependency, being labor-consuming and time-intensive, and requiring *a priori* choice of regions to be investigated.

To overcome these shortcomings, another automated method of measuring brain atrophy has been developed [97,98,99,100]. The method of voxel-based morphometry (VBM) objectively maps gray matter loss on a voxel-by-voxel basis after anatomical standardization analogous to that used in functional neuroimaging. The advantage of VBM over analyses based on ROI analysis is that VBM produces an unbiased result from exploration of the whole brain. Testa et al. reported higher accuracy of discriminating AD and controls than ROI-based analysis [101]. One of these popular statistical tools based on the voxel is statistical parametric mapping (SPM) that refers to the construction and assessment of spatially extended statistical processes used to test hypotheses about functional imaging data (http://www.fil.ion.ucl.ac.uk/spm/, accessed on 29 March 2021). Statistical parametric mapping (SPM) [102], a univariate hypothesis-driven method, provides simple and computationally efficient approaches to produce maps of task-related activations with estimates of their levels of significance based on a statistical parameter of each voxel. However, it results in a loss of sensitivity if the fMRI experiment induces co-activation of spatially disparate areas with slightly different temporal behaviors. 

Unlike univariate methods, multivariate data-driven techniques enable an exploratory analysis of fMRI datasets and may potentially separate meaningful activation by computing suitable statistical models independent of nay reference paradigm [103]. Furthermore, multivariate nature exploits the relationship between voxels and may possibly provide useful information about co-activation in spatially different areas of the brain. In general, multivariate analysis might have increased sensitivity compared to univariate analysis even when the disease-related changes in cerebral blood flow (CBF) originate in clearly circumscribed foci and spread spatially during the disease course. Multivariate analysis can detect these subtle, but robust changes, although univariate analysis might experience overly stringent false-positive corrections that tend to ‘correct away’ the true effects (as evidenced by the results of our voxel-wise analysis).

Habeck et al. reported that multivariate analysis might be more sensitive than univariate analysis for the early diagnosis of AD [44]. The multivariate techniques do not necessarily rely on underlying “networks” of pathology. Asllani et al. used multivariate approaches to evaluate correlation/covariance of CBF measurement across brain regions rather than proceeding on a region-by-region (or voxel-by-voxel) basis [104].

Among the multivariate data-driven techniques, ICA/BSS has been shown to provide a powerful method for the exploratory analysis of fMRI data [79,105,106,107]. ICA does not require the specification of temporal signal profiles or anatomical ROIs to generate meaningful spatiotemporal patterns of brain activity [108].

The multivariate statistical nature of ICA allows one to transform three-dimensional fMRI data sets into brain activity patterns starting from the spatial or temporal covariance of the measured signals and reveals multiple spatiotemporal ‘modes’ of signal variability [11]. This transformation is achieved by imposing the general, yet neurophysiologically plausible, constraint of removing the statistical dependence of the output modes [105]. To meet this constraint, the value distribution of the fMRI signals in space or time is to be considered: two variants called sICA and tICA. The former refers to the statistical distribution of signals across the sampled hemodynamic locations, while the latter refers to the statistical distribution of source signals across the sampled time-points [82].

### 6.2. Comparison of ICA/BSS Algorithms

As there are a variety of ICA algorithms, it is also important to compare their performance to better understand their strengths and limitations [108].

Correa et al. [91] had compared performances of five algorithms included in the toolbox GIFT: Informax [64], FastICA [67], Joint Approximate Diagonalization of Eigen Matrices (JADE) [109], Simultaneous Blind Extraction using Cumulants SIMBEC [110], and Algorithm for Multiple Unknown Signal Extraction (AMUSE) [4]. Based on their results of fMRI data, Informax emerged as a reliable choice for the task, followed by JADE as a close second. FastICA performed reliably for most cases as well whereas the performance of SMIBEC and AMUSE did not prove to be robust. SIMBEC may prove to be useful to identify sub-Gaussian sources. The performance of AMUSE is highly dependent on the differentiability of the spectra of the sources.

### 6.3. Spatial, Temporal, and Spatiotemporal ICA

ICA is a technique that attempts to separate data into maximally independent groups, achieving maximal independence in space or time to yield three varieties of ICA meaningful for fMRI applications: spatial ICA (sICA), temporal ICA (tICA), and spatiotemporal ICA.

Since the first application of ICA for fMRI analysis [105], it has been controversial to choose spatial or temporal independency. McKeown et al. argued on the sparsely distributed nature of the spatial pattern with sICA. The applications of temporal ICA to fMRI data have appeared [74,82]; however, spatial ICA has by far dominated the functional imaging literature to date. The most important reasons for this are that the spatial dimension is much larger than that of the temporal dimension in fMRI data [82].

Stone et al. proposed a method that attempts to maximize both spatial and temporal independence [83]. Seifritz et al. presented an interesting combination of sICA and tICA [111] and used an initial sICA to reduce the spatial dimensionality of the data by locating a ROI in which they then performed tICA to study in more detail the structure of the nontrivial temporal response in the human auditory cortex.

### 6.4. How Many Components Are There?

Before applying ICA/BSS to fMRI, characteristics of the independent components must be determined.

In general, fMRI data may be composed of signals of interest and signals not of interest (noises). Signals of interest include task-related, function-related, and transiently task-related signals [80]. Signals not of interest include physiology-related (breathing and heart rate), motion-related (mouth movement in the naming task), and scanner-related (scanner drift and system noise, susceptibility, and radio frequency artifacts) signals. In addition, there are several types of noise in an fMRI experiment, such as object variability, thermal noise, patient movement, brain movement, and so on. In the ICA model, these noises are often not explicitly modeled, but rather manifested as separate components [105,112].

Another problem is determining the number of components. Beckmann et al. and Calhoun et al. used different methods to estimate the number of components in fMRI data [87,113]. McKeown et al. [105] applied ICA to fMRI data and calculated the contribution of each component to fMRI data for extracting consistently task-related, transiently task-related, slowly varying, quasiperiodic, movement-related, and residual noise components. McKeown et al. also applied a combined PCA/ICA approach to estimate the number of spatially independent components contained in fMRI data [79]. Calhoun et al. used standard information theoretic methods for estimating the number of components from the aggregate data set [113]. The number of sources can be estimated using Akaike’s information criterion or the minimum description length criterion [114,115].

### 6.5. Application of ICA/BSS to AD Diagnosis

Many applications of ICA/BSS to brain signal processing exist, such as removal artifact from EEG [14] and MEG [77,78], analysis of evoked magnetic fields [14,116,117], fMRI data [90], and in clinical research, for example the diagnosis of AD [118]. In this section, we will focus on application of ICA/BSS to the diagnosis of AD.

Chapman et al. [119] used PCA to identify and measure the ERP components. Their scores to relevant and irrelevant stimuli were used in discriminant analyses to develop functions that successfully classified individuals as belonging to an early-stage AD group or a like-aged control group, with probabilities of an individual belonging to each group. Additionally, 92% of the subjects were correctly classified into either the AD group or the control group with a sensitivity of 1.0. Besthorn et al. [120] used PCA as a postprocessing tool for compressing linear and nonlinear EEG features over channels. They obtained 95.9% correct classification using age as a moderator variable in the study.

A number of EEG studies [4,121] on AD and MCI have reported several typical findings such as a slowing and diffusing of the posterior dominant alpha activity, an unclear alpha attenuation after eye-opening, as well as an increase of delta and theta, and a decrease of beta and gamma activities [5]. For more details, see a review of signal processing techniques applied for revealing pathological changes in EEG associated with AD [5]. Cichocki et al. investigated the application of ICA/BSS methods as preprocessing tools with possible application for AD diagnosis. They propose an approach of filtering EEG data based on ICA/BSS that can significantly improve the sensitivity and specificity of EEG-based diagnosis of AD at the early stage [4]. The team employed a non-ICA based method of BSS: the algorithm for multiple unknown signals extraction (AMUSE [122]), on data from MCI patients who progressed to AD and age matched controls, achieving a classification rate of 80% for MCI based on linear discriminant analysis (LDA) [4].

Vialatte et al. explored the group differences between Mild AD patients and control subjects, finding that precleaning data with ICA using the improved weight-adjusted second-order blind identification (IWASOBI) algorithm amplified the differences between the groups [123]. Melissant et al. used ICA to reduce artifacts of EEG data to improve classification results for patients in an initial stage [121]. They made a conclusion that a more robust detection of AD-related EEG patterns may be obtained by employing ICA as ICA-based preprocessing of EEG data can improve classification results for AD patients in an initial stage. Jervis et al. applied ICA and cluster analysis to EEG P300 data obtained from healthy and AD subjects [27] showed that the latencies of the back-projected independent components (BICs) of the P300 differed between healthy participants and AD patients. They proposed that the latencies of the BIC associated with the P3b component may be a suitable biomarker for AD.

Escudero et al. [124] analyzed MEG background activity recordings acquired with a 148-channel whole-head magnetometer from 21 AD patients and 21 control subjects using the algorithm for multiple unknown signals extraction (AMUSE [4]) to blindly decompose artifact-free epochs of 20s. Their preliminary results showed that the proposed procedure based on BSS and selection of significant components may improve the classification of AD patients using straightforward features from MEG recordings. Fernandez et al. [125] applied PCA to the mean frequency from the MEG signals of 22 patients with AD, 22 patients with MCI, and 21 healthy controls. Results demonstrated the mean frequency score seem to be adequate and sensitive to detect differences between normal aging, cognitive deterioration, and AD.

Higdon et al. [9] applied PCA to FDG-PET for the diagnosis of AD from fronto-temporal dementia and reported slightly better results with images preprocessed with principal least squares analysis than with PCA when using a classifier based on linear discriminant analysis Habeck et al. [44] examined the efficacy of multivariate and univariate analytic methods for the diagnosis of early AD. Using the extended PCA-approach to the FDG-PET data from two clinical populations, they analyzed the spatially correlated metabolism as a function of AD status and reported that the multivariate marker’s diagnostic performance in the replication samples was superior to that of the univariate marker’s. Kerrouche et al. first applied a novel voxel-based multivariate technique, such as PCA, to a large FDG-PET data set to investigate whether it is possible to distinguish vascular dementia from AD. They used PCA to remove PCs significantly correlated to age. Their results show the potential of voxel-based multivariate methods to highlight independent functional networks in dementing disease. By maximizing the separation between groups, this method extracted a metabolic pattern that efficiently differentiated vascular dementia and AD [126].

Chen et al. proposed a simple and automated method for the measurement of changes in brain volume from an individual’s sequential MRIs using an iterative PCA (IPCA) [127]. The IPCA considered the voxel intensity pairs from coregistered MRIs and identified those pairs a sufficiently large distance away from the iteratively determined PCA major axis. Their results demonstrated IPCA’s ability to characterize whole-brain atrophy rates in patients with AD [127,128].

Su et al. [12] presented the Hybrid wavelet-ICA to investigate the use of ICA dynamic PET data both in the image domain and in the wavelet domain, where the data had been transformed using Battle-Lemarie wavelets, as in the article [129].

Greicius et al. adapted ICA to derive the default-mode network [130,131] in a more data-driven fashion (i.e., without requiring a priori specification of a seed region). Examination of the default-mode network in these groups revealed three critical findings: a significant coactivation of the hippocampus in the default-mode network, the network is abnormal in the mildest stages of AD compared to healthy aging, and network activity holds potential as a non-invasive biomarker of incipient AD [131].

By using resting-state fMRI and combining correlation and ICA, Tang [132] explored a new method for resting-state functional connectivity [133,134] that was finally applied to seek the abnormality of the brain functional network under the pathophysiology of AD.

Celone et al. [118] used ICA to investigate memory-related fMRI activity in 52 individuals across the continuum of normal aging, MCI, and mild AD. Memory function is likely subserved by multiple distributed neural networks that are disrupted by the pathophysiological processes of AD. ICA revealed specific memory-related networks that activated or deactivated during an associative memory paradigm. Across all subjects, hippocampal activation and parietal deactivation demonstrated a strong reciprocal relationship. Less impaired MCI subjects showed paradoxical hyperactivation in the hippocampus compared with controls, whereas more impaired MCI subjects demonstrated significant hypoactivation, similar to the levels observed in the mild AD subjects.

Sorg et al. combined ICA and ROI-based correlation methods to investigate resting-state networks (RSNs) in patients with MCI [90]. They analyzed fMRI and structural MRI data from healthy elderly and patients with amnestic MCI, a syndrome of high risk for developing AD and concluded that, in individuals at risk for AD, a specific subset of RSNs is altered.

Rombouts et al. applied tensor PICA [15] to study fMRI signal during face encoding in 18 AD, 28 MCI patients, and 41 healthy elderly controls [135]. The tensor PICA showed activation in regions associated with motor, visual, and cognitive processing, and deactivation in the default mode network. They concluded that the tensor PICA is a promising tool to identify and detect differences in (de)activated brain networks in elderly controls and dementia patients.

### 6.6. ICA as a Component of Machine Learning Models

Computer aided diagnosis (CAD) is poised to be a considerable tool for identifying cases of Alzheimer’s disease and a host of other diseases. CAD typically involves the application of a machine learning (ML) model onto features derived from neuroimaging, biomarkers, and others. ML can be used in tandem with expert review; in cases of MRI and other image analysis, for example, a ML model can ascertain details and features that are difficult for a human to identify by eye. Support vector machines, as well as other model types such as decision trees and neural networks have been developed to diagnose Alzheimer’s disease based on MRI, PET, and other imaging methods with high accuracy [136,137,138,139,140,141]. Models have also been effective when trained on EEG feature data as well [142,143,144,145,146].

ICA, which can be used for signal separation and feature extraction, can be implemented as a component of a machine learning-based diagnostic system. ICA can be employed to initially extract and transform features of a dataset, which can then be fed into a machine learning model. Such systems have been shown to be more effective that direct ML in problems such as facial recognition [147] and classification based on microarray data [148].

Such methods can be similarly applied to applications of brain health and imaging. Artificial neural networks and support vector machines (SVMs) working with data preprocessed with ICA have been found to detect artifacts in EEGs with an accuracy/alignments between 89.13% and 95.20% compared to expert rating [149]. The non-ICA based algorithm AMUSE was used y Vialatte et al., to process data for input for sparse-bump modeling, which was fed into a neural network geared towards the classification of MCI cases (that later progressed to AD), achieving a 93% classifier rate [150], an improvement from the aforementioned LDA with an 80% implemented by the team [4]. Data suggests that perhaps ICA based methods are superior to some standard BSS methods for artifact removal. Cassani et al. tested statistical artifact rejection (SAR), blind source separation based on second order blind identification and canonical correlation analysis (BSS–SOBI–CCA), and wavelet-enhanced independent component analysis; the removal of artifacts by experts was also assessed as a standard. The resulting features were used to train an SVM based classifier. The ICA-model performed comparably to the standard in discriminating normal and mild AD, and slightly better than the standard in discriminating mild and moderate AD. BSS was generally inferior to ICA and SAR, possibly due to the removal of discriminatory data [151]. Other specific analyses of brain development and health have been examined with similar combined-methods [152,153,154,155,156].

With regard to Alzheimer’s disease, CAD systems incorporating ICA-processed into a machine learning model have been developed with some success. Khedher et al. developed a SVM based CAD system trained with MRI data from the Alzheimer’s disease neuroimaging initiative (ADNI) that was able to distinguish between cognitively normal individuals and those with MCI or AD that notably featured graphical representations of the input features that makes the basis of the system’s decision making more clear. The system had 79% accuracy distinguishing MCI from NC, 89% accuracy distinguishing AD from normal controls (NC), and 85% accuracy distinguishing MCI from AD cases [157].

Yang et al. applied similar methods to data from open access MRI datasets such as the ADNI; features processed and extracted with FastICA [158] were applied to a SVM classifier. Applying the system to grey-matter only images achieved an accuracy of approximately 89% when distinguishing between NC and AD subjects, and 81% between NC and MCI subjects at a 90–10 training/testing split; whole-brain image performance was notably lower [159]. A later 2017 model by Yang et al., using two stage component number estimation and ICA combined with clinical features used as inputs for a SVM, achieved an accuracy of 97.7% and 87.8% in distinguishing AD and MCI from NC, respectively [160].

Qiao et al., developed a three-level hierarchical partner matching ICA method; functional MRI data was first processed with spatial ICA and group-mapping of IC groups, followed by partner matching group-map clusters and cluster-map generation, then partner mapping of cluster-maps. Tracing the optimal clusters derived from the cluster-maps backwards indicates the most stable ICs. Inputs were fed into a directed acyclic graph neural network incorporating convolutional layers; an accuracy of 95.59% was achieved in leave-one-out cross validation [161].

Basheera et al. used MRI grey matter images segmented with hybrid enhanced independent component analysis applied to a convolutional neural network. Training was done with ADNI image data and, on a sample of 21 independent MRI slices, the output of the network was compared to the decision of a physician, and an accuracy of 90.47% was achieved [162].

Other imaging methods beyond MRI have been used as well. Illán et al. conducted a comparison of SVM based CAD systems based on voxels-as-features, principal component analysis, and ICA processing on SPECT images. Models based on samples with NC and AD patients either grouped (method 1) or split into 3 subgroups based on disease severity/characteristics (method 2) were tested. Accuracies of PCA and ICA were similar for both method 1 (88.61% and 89.87%, respectively) and method 2 (88.61% and 91.14%, respectively) and both were greater than the VAF baseline (72.15% and 74.68% for method 1 and method 2, respectively) [136]. Toussaint et al. utilized spatial ICA of fluoro-deoxygenase (FDG) PET images, combined with other clinical features such as cognitive test scores. Good accuracies were noted in leave-one-out cross validation distinguishing NC from preclinical AD, but lower when attempting to distinguish between stable and converting MCI [163].

## 7. Conclusions

AD is a progressive neurodegenerative disorder and the most common cause of dementia associated with aging. The diagnosis of AD remains largely based upon clinical assessment, and is often made at relatively late stages of the pathophysiological process. As disease-modifying therapies are likely to be most efficacious at much earlier stages of the disease, it is important to develop markers for early disease detection in individuals who are at risk for AD. Fortunately, some biomedical techniques such as EEG, CT, PET, MRI, and fMRI, can non-invasively acquire brain signals to aid in a more objective assessment of AD pathology, even when AD is at an early stage. As mentioned above, we have introduced the applications of these biomedical techniques as potential AD diagnostic tools.

The brain signals sampled from individuals using these biomedical techniques are the mixture of many signals of interest or non-interest; collectively, AD-related signals or noises serve different purposes for analysis. A key challenge is to acquire useful AD-related signals and to discover biomarkers from the sampled brain signals. To address this challenge, many novel signal processing techniques have been developed. In this paper, we focused on reviewing applications of ICA/BSS approaches to the diagnosis of AD.

ICA/BSS is one of the data-driven, multivariate, and unsupervised methods without any *a priori* information. The quite fruitful applications of ICA/BSS to brain signals have shown that such technique is very useful and powerful. Its ability to represent the high-dimensional data, especially MRI or fMRI data, enables it to be a powerful tool for clinical AD neuroimaging biomarker discovery.

## Data Availability

Not applicable.

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
