# Peer review of "An Overview of ICA/BSS-Based Application to Alzheimer’s Brain Signal Processing"

_biomedicines, 2021, doi:10.3390/biomedicines9040386_

Round 1
Reviewer 1 Report
The article is an overview focused on the usefulness of Independent Component Analysis or Blind Source Separation (ICA/BSS) in Alzheimer's disease (AD). This research in data mining is very interesting in the search for biomarkers, given objective measures are needed to make it easier and more accurate the early diagnosis of AD. The issue addressed in the manuscript is interesting from the perspective of brain signal analysis.
The review covers the usages of ICA/BSS applied to different diagnostic techniques. In general, the manuscript is clear and achieves the objectives. However, some points should be revised to improve the clarity. Detailed comments are listed below:
3. Biomedical techniques for detecting Alzheimer’s brain signals
In this section, a very general finding in EEG results is described:
"The primary EEG marker of AD is slowing of the
rhythms and a decrease in coherence among different brain regions". However, in other techniques, the results found are explained in more detail. This gives an appearance of imbalance. The Authors should explain in more detail the main findings in EEG and the contribution of ICA/BSS processing.
4. THEORY AND MODEL OF ICA/BSS
- This section would be more readable to a non-specialized public by first introducing the general principles behind ICA/BSS
An example:
https://www.researchgate.net/publication/265973497_Blind_source_separation_and_ICA_techniques_a_review
- When explaining the ICA subtypes, in the text it remains unclear whether the studies are referred to AD or not.
- Also, a figure showing an example of the effects of applying ICA/BSS to different brain signals could be useful for visually understanding their usefulness.
- Please note that there is some text after the skew-pdf formula (w h from "where")
6. APPLICATION OF ICA/BSS TO AD DIAGNOSIS
The authors should explain in more detail the results based on ICA processing of EEG data for AD diagnosis.
In general, in the overview, there is much more information reviewed about the usefulness of ICA for image than electrophysiological techniques. I ignore whether it corresponds to a lack in the bibliography about the usage of ICA applied to EEG in AD but it leads to reading it as a biased compilation. EEG usages of ICA in AD should be explained in more detail.
In the case of EEG preprocessing, for me, there is a theoretical problem to solve. The slow waves are traveling waves, that is, they are generated in several brain regions and then travel across the entire cortex and many subcortical structures. Thus, the same wave can be recorded from many distributed regions. An therefore, ICA preprocessing can "erase" them of the signal given they are common to all channels. However, in some conditions, these waves are relevant to understand the state of the network. For example, AD patients have less slow-wave sleep, which is physiologically relevant since clearance of toxic metabolites happens locked with this slow waves. So, it is interesting not losing them. Could the Authors shed any light on this?
7. ICA/BSS APPLICATIONS TO BRAIN SIGNAL PROCESSING FOR AD
DIAGNOSIS
- The section title is very similar to the previous section. I recommend choosing a more descriptive title to differentiate the contents of each one.
- Akaike's information criterion: there is a mistake in the name
8. CONCLUSION
Alzheimer’s disease should be abbreviated
Author Response
Reviewer 1’s comments and authors’ responses
The article is an overview focused on the usefulness of Independent Component Analysis or Blind Source Separation (ICA/BSS) in Alzheimer's disease (AD). This research in data mining is very interesting in the search for biomarkers, given objective measures are needed to make it easier and more accurate the early diagnosis of AD. The issue addressed in the manuscript is interesting from the perspective of brain signal analysis.
The review covers the usages of ICA/BSS applied to different diagnostic techniques. In general, the manuscript is clear and achieves the objectives. However, some points should be revised to improve the clarity. Detailed comments are listed below:
- Biomedical techniques for detecting Alzheimer’s brain signals
In this section, a very general finding in EEG results is described:
"The primary EEG marker of AD is slowing of the
rhythms and a decrease in coherence among different brain regions". However, in other techniques, the results found are explained in more detail. This gives an appearance of imbalance. The Authors should explain in more detail the main findings in EEG and the contribution of ICA/BSS processing.
Response: We have added more detailed information about the principal EEG findings in AD cases, though we have kept this brief to be more in-line with the other methods discussed within the section..
- THEORY AND MODEL OF ICA/BSS
- This section would be more readable to a non-specialized public by first introducing the general principles behind ICA/BSS
An example:
https://www.researchgate.net/publication/265973497_Blind_source_separation_and_ICA_techniques_a_review
- When explaining the ICA subtypes, in the text it remains unclear whether the studies are referred to AD or not.
Response: The studies referenced in the ICA subtype sections relate directly to the technique described in most cases, rather than a specific neurological disorder such as AD. We have added a note at the beginning of the section to make this more clear.
- Also, a figure showing an example of the effects of applying ICA/BSS to different brain signals could be useful for visually understanding their usefulness.
Response: Though figures would be useful, we believe they would be easy enough to find for an interested reader in a variety of contexts, and the manuscript is rather long as it is.
- Please note that there is some text after the skew-pdf formula (w h from "where")
Response: We have adjusted the text-wrapping to fix this error. Thank you for pointing it out.
- APPLICATION OF ICA/BSS TO AD DIAGNOSIS
The authors should explain in more detail the results based on ICA processing of EEG data for AD diagnosis.
In general, in the overview, there is much more information reviewed about the usefulness of ICA for image than electrophysiological techniques. I ignore whether it corresponds to a lack in the bibliography about the usage of ICA applied to EEG in AD but it leads to reading it as a biased compilation. EEG usages of ICA in AD should be explained in more detail.
In the case of EEG preprocessing, for me, there is a theoretical problem to solve. The slow waves are traveling waves, that is, they are generated in several brain regions and then travel across the entire cortex and many subcortical structures. Thus, the same wave can be recorded from many distributed regions. An therefore, ICA preprocessing can "erase" them of the signal given they are common to all channels. However, in some conditions, these waves are relevant to understand the state of the network. For example, AD patients have less slow-wave sleep, which is physiologically relevant since clearance of toxic metabolites happens locked with this slow waves. So, it is interesting not losing them. Could the Authors shed any light on this?
Response: We have bolstered the sections regarding EEG in sections 6.5 and 6.6. Removal of artifacts in any medium can result in a loss of information, and that is a trade-off that should be considered. Many ICA methodologies purport that this loss is minimal, and general increases in performance such as ML model classification lends credence to the idea that it is generally worthwhile for the purpose of a diagnostic. The specific method (if any) of ICA or BSS to apply depends on what is being looked at, and what sorts of assumptions need to be made.
- ICA/BSS APPLICATIONS TO BRAIN SIGNAL PROCESSING FOR AD
DIAGNOSIS
- The section title is very similar to the previous section. I recommend choosing a more descriptive title to differentiate the contents of each one.
Response: We have opted to restructure section 7 (now section 6) to include the original section 6 (now section 6.5).
- Akaike's information criterion: there is a mistake in the name
Response We have corrected the misspelling, thank you for pointing it out.
- CONCLUSION
Alzheimer’s disease should be abbreviated
Response: We have corrected this, thank you for pointing it out.
Reviewer 2 Report
Independent component analysis as a specific embodiment of blind source separation (ICA/BSS) algorithm based on time, frequency and joint time-frequency decomposition of data is becoming an increasingly popular tool for analyzing biomedical data with potential applications in many areas from applied sciences like neuroscience. In the current study the authors have focused on the ICA/BSS as an important signal processing approach in the early diagnosis of Alzheimer's disease. The study is interesting and the paper well organized but the information per se is complicated, difficult to understand and confusing for scientists not working in the field; i.e physicians are not familiar with these models (not even students of the early years of electrical engineering). Thus, a) a table with the advantages and dis-advantages of the reported univariate or multivariate techniques for analyzing MRI datasets and a) table showing the importance of ICA and extensions or ICA/BSS variants in relation to fMRI/MRI or other imaging techniques (i.e , EEG, MEG, PET, etc) in order to distinguish the useful AD-related signals, would be of help
Author Response
Reviewer 2’s comments and authors’ responses
Independent component analysis as a specific embodiment of blind source separation (ICA/BSS) algorithm based on time, frequency and joint time-frequency decomposition of data is becoming an increasingly popular tool for analyzing biomedical data with potential applications in many areas from applied sciences like neuroscience. In the current study the authors have focused on the ICA/BSS as an important signal processing approach in the early diagnosis of Alzheimer's disease. The study is interesting and the paper well organized but the information per se is complicated, difficult to understand and confusing for scientists not working in the field; i.e physicians are not familiar with these models (not even students of the early years of electrical engineering). Thus, a) a table with the advantages and dis-advantages of the reported univariate or multivariate techniques for analyzing MRI datasets and a) table showing the importance of ICA and extensions or ICA/BSS variants in relation to fMRI/MRI or other imaging techniques (i.e , EEG, MEG, PET, etc) in order to distinguish the useful AD-related signals, would be of help
Response: Although we agree that a table might be useful, the multitude of methods used and the different circumstances that might drive selection (E.g., data type, presence of temporal and/or spatial dimensions, etc.), we believe that a table might end up large and convoluted given the explanation necessary. We acknowledge that our explanation of the basics was lacking, and we have tried to rectify this by adding some information and pointing the reader towards more thorough reviews of the basic concepts, as well as defining certain vocabulary that may not be commonly known.